# Development of the Nursing Nutritional Care Behaviors Scale (B-NNC) in Italian and Psychometric Validation of Its German Translation in Austria

**DOI:** 10.3390/nursrep15050146

**Published:** 2025-04-28

**Authors:** Rosario Caruso, Loris Bonetti, Silvia Belloni, Cristina Arrigoni, Arianna Magon, Gianluca Conte, Valentina Tommasi, Silvia Cilluffo, Maura Lusignani, Stefano Terzoni, Silvia Bauer

**Affiliations:** 1Health Professions Research and Development Unit, IRCCS Policlinico San Donato, 20097 San Donato Milanese, Italy; arianna.magon@grupposandonato.it (A.M.); gianluca.conte@grupposandonato.it (G.C.); 2Department of Biomedical Sciences for Health, University of Milan, 20133 Milan, Italy; silvia.cilluffo@unimi.it (S.C.); maura.lusignani@unimi.it (M.L.); stefano.terzoni@unimi.it (S.T.); 3Department of Business Economics, Health and Social Care, University of Applied Sciences and Arts of Southern Switzerland (SUPSI), 6928 Manno, Switzerland; loris.bonetti@supsi.ch; 4Nursing Research Competence Centre, Department of Nursing, Ente Ospedaliero Cantonale, 6500 Bellinzona, Switzerland; 5Department of Public Health, Experimental and Forensic Medicine, Section of Hygiene, University of Pavia, 27100 Pavia, Italy; cristina.arrigoni@unipv.it; 6Bachelor School of Nursing, ASST Grande Ospedale Metropolitano Niguarda, 20162 Milan, Italy; valentina.tommasi@ospedaleniguarda.it; 7Institute of Nursing Science, Medical University of Graz, Neue Stiftingtalstraße 6/P06-WEST, 8010 Graz, Austria; silvia.bauer@medunigraz.at

**Keywords:** malnutrition, older adults, nursing behaviors, nutritional care, psychometric validation, questionnaire development

## Abstract

**Background/Objectives**: Malnutrition among older adults remains a significant healthcare issue, yet existing questionnaires primarily measure knowledge and attitudes rather than actual behaviors. This study aimed to develop the Nursing Nutritional Care Behaviors Scale (B-NNC Scale) in its original Italian version, translate it into German, and evaluate its psychometric properties in registered nurses and nurse assistants in Austria. **Methods**: This study followed a two-phase, multi-method design. In Phase 1 (Development Phase), the scale was developed in Italian through a scoping review, expert focus group, and content validation involving 18 clinical nutrition experts using the Content Validity Ratio (CVR). In Phase 2 (Validation Phase), the scale was translated into German through a cross-cultural adaptation process, pilot-tested, and psychometrically validated in a sample of 1072 nurses and nurse assistants working in Austrian hospitals across various clinical settings. Exploratory and confirmatory factor analyses (EFA and CFA) were performed to assess construct validity, measurement invariance between professional roles was tested, and internal consistency was measured using McDonald’s Omega. **Results**: Content validity was confirmed with a mean CVR of 0.634. EFA suggested a three-factor solution—(1) Nutritional Assessment and Calculation Skills, (2) Nutritional Evaluation and Care Planning, and (3) Nutritional Support and Care Implementation—leading to the retention of 19 items. CFA supported this structure, and McDonald’s Omega indicated high internal consistency across subgroups. Partial measurement invariance revealed some differences in response patterns between registered nurses and nurse assistants. **Conclusions**: The B-NNC Scale demonstrated robust validity and reliability in measuring self-reported nursing behaviors related to nutritional care in older adults. It addresses a notable gap in existing instruments and may serve as a valuable tool for research and practice to improve malnutrition management.

## 1. Introduction

Malnutrition is a prevalent and serious issue among older adults, leading to significant health consequences, including increased morbidity, prolonged hospital stays, and higher mortality rates [1,2]. Several international guidelines provide recommendations for effectively managing malnutrition in this population [3,4,5]; however, adherence to these guidelines remains suboptimal in clinical practice [6,7,8]. Improving adherence requires a deeper understanding of the factors influencing nursing practice, particularly the knowledge, attitudes, and behaviors of nursing staff regarding nutritional care.

Assessing knowledge, attitudes, and behaviors is critical, as these factors directly influence clinical decision-making and the implementation of nutritional interventions [9,10,11]. One widely used approach to evaluating these dimensions is through Knowledge, Attitude, and Practice (KAP) questionnaires, which assess what is “known, believed, and done” about a specific health issue [12,13,14,15,16,17,18]. Such tools provide essential insights for designing targeted educational interventions that enhance professional competencies and promote evidence-based care [12,13,14,15,16,17,18].

Despite the availability of instruments assessing knowledge and attitudes regarding nutritional care in older adults—such as the Knowledge of Malnutrition-Geriatric (KoM-G) 2.0 questionnaire [19,20] and the Staff Attitudes to Nutritional Nursing Care Geriatric (SANN-G) scale [20,21,22,23]—there is a critical gap in tools assessing nursing behaviors related to nutritional care. Currently, no validated instrument systematically captures nurses’ self-reported behaviors, reflecting how nutritional care is delivered in practice.

This gap has significant implications for both research and clinical practice [24]. Without a validated measure of behaviors, it is challenging to determine whether nurses’ clinical behaviors align with best practices in malnutrition management [25]. Research on behavioral determinants of nutritional care remains incomplete, limiting the ability to design interventions that effectively translate knowledge and attitudes into practice. In clinical settings, the lack of a behavioral assessment tool makes it difficult to identify areas for improvement, monitor changes over time, or evaluate the impact of training programs [26]. Addressing this gap is essential to ensure that nutritional care is understood, valued, and consistently applied in daily nursing practice [27].

In light of this global gap, it is crucial to clarify why Italy and Austria were chosen as initial contexts for developing and validating this scale. Previous research demonstrated difficulties in effectively integrating nutritional care best practices within hospital environments for both countries [6,22]. Research in Austria has revealed notable shortcomings in structural and procedural indicators concerning nutritional care delivered by nursing staff [6]. Similarly, studies conducted in Italy have identified gaps in nurses’ nutritional knowledge and highlighted inadequate incorporation of nutritional care into daily nursing routines [22,27]. Although existing studies in these national contexts have investigated nurses’ knowledge, attitudes, and self-efficacy, no current instrument specifically measures nurses’ actual behaviors, representing concrete actions demonstrating adherence to clinical nutritional guidelines.

Given the increasing demand for cross-cultural research, it is also essential that newly developed instruments undergo thorough translation and psychometric validation processes to confirm their applicability across diverse healthcare settings [28]. A robust validation process, including reliability and validity testing, is essential to confirm that a new behavioral assessment tool functions equivalently across linguistic and cultural settings [29]. Recognizing the absence of a self-report instrument for assessing nurses’ clinical behaviors regarding nutritional care, this study specifically aims to develop the Nursing Nutritional Care Behaviors Scale (B-NNC Scale) in its original Italian version, translate it into German, and validate its psychometric properties among registered nurses and nurse assistants in Austria. The B-NNC Scale will serve as a structured, evidence-based instrument to comprehensively evaluate nursing behaviors associated with nutritional care, which could support research initiatives and clinical practice. This study leverages existing evidence and collaborative networks in these two settings to facilitate robust scale development and validation processes. Selecting Italy and Austria provided an opportunity for context-specific yet methodologically rigorous research supported by sufficient sampling strategies and bilingual research expertise.

## 2. Materials and Methods

### 2.1. Design

This study employed a multi-phase, multi-method design, adhering to best practices for developing and validating new self-reported measures [30,31]. The study was conducted in two distinct phases: (a) Phase 1: Conceptualization Phase, and (b) Phase 2: Validation Phase. The Ethical Committee of Ospedale San Raffaele approved this study (prot. N. 74/INT/2022) in June 2022. The study was also approved by the Medical University of Graz (EK-Nr.: 35-112 ex 22/23) in January 2023.

Phase 1 aimed to conceptualize the B-NNC Scale through three primary steps. First, a comprehensive literature review was conducted to identify the main challenges related to nutritional care behaviors performed by registered nurses and nurse assistants. Second, findings from the review were discussed with the Italian research team during a focus group session to refine the understanding of these challenges in practice. Third, an initial pool of items for the scale was generated through a panel discussion among research team members in Italy, defining the a priori dimensions of the scale.

Phase 2 was conducted in two countries, Italy and Austria. In Italy, the content validity of the scale was assessed. In Austria, a translation and adaptation process was performed, including forward translation, back-translation, and consensus meetings to ensure cultural and linguistic equivalence [32,33]. A pilot test with Austrian nurses and nurse assistants was conducted to refine the translated version, ensuring the understandability of the items. Lastly, a cross-sectional study was carried out in Austria to collect data for the psychometric validation of the scale, including its reliability and construct validity. In both countries, participants were recruited using predefined eligibility criteria. For the content validity phase in Italy, inclusion criteria required participants to be clinical nutrition experts with (a) at least five years of professional experience, (b) demonstrated expertise in nutritional care for older adults in either clinical or academic settings, and (c) advanced academic qualifications (e.g., MSN, PhD). Experts not proficient in Italian were excluded, as the item pool was originally developed in Italian (see Phase 2a). For the Austrian pilot testing and cross-sectional validation, the inclusion criteria were (a) being a registered nurse or nurse assistant, (b) currently employed in an Austrian hospital, and (c) having at least six months of clinical work experience. Exclusion criteria included: (a) lacking direct patient care responsibilities (e.g., administrative or academic-only roles), and (b) being a nursing student or trainee without independent clinical duties.

#### Development and Validation Settings

This study was collaboratively designed by a research team composed of Italian and Austrian scholars. The original development of the B-NNC Scale was carried out in Italian for pragmatic reasons: most authors were Italian-speaking and affiliated with institutions in Italy, which facilitated the coordination of item generation activities, expert panel recruitment, and content validation processes. Consequently, the scale was initially developed in this language (see Phase 1) and later translated into German for further validation, paving the way for future possible validations in other languages.

The Italian version of the B-NNC Scale underwent a thorough content validation process involving Italian-speaking experts in clinical nutrition (see Phase 2: Content validity of the Italian version), confirming item relevance and clarity. However, due to limited access to a sufficiently large and diverse sample in Italy at the time of data collection, full psychometric validation (e.g., exploratory and confirmatory factor analysis, internal consistency, and measurement invariance) could not be conducted in Italian in this stage. Data collection for this purpose is ongoing for future scaling up of the research project.

Austria was selected for the second phase of the study, which involved translating the B-NNC Scale into German and conducting psychometric validation. This choice was based on the availability of institutional partners and healthcare settings capable of recruiting a sample large enough to meet methodological requirements for cross-sectional validation and factorial analyses. The Austrian phase, therefore, allowed for a comprehensive assessment of the scale’s construct validity and reliability. Future studies are warranted to address this gap.

### 2.2. Conceptualization Phase (Phase 1)

#### 2.2.1. Phase 1a: Literature Review

This step was conducted as a scoping review to address the question: “What are the challenges, components, and practices related to nutritional care behaviors for older adults performed by registered nurses and nurse assistants, as reported in the published literature?” The review followed best practices for scoping reviews and was performed by two authors (SB and GC) with expertise in this methodology [34,35].

The Population, Concept, and Context (PCC) framework guided the review [35], where the population included registered nurses and nurse assistants, the concept focused on nutritional care behaviors for older adults, and the context encompassed published literature. The databases searched included PubMed, CINAHL, Web of Science (WoS), and Scopus, with search queries detailed in Table 1. The final search was conducted in November 2022 and was based on a previous literature review [27].

The inclusion and exclusion criteria were pre-defined to ensure the relevance and focus of the review. Articles addressing nutritional care behaviors for older adults specific to registered nurses and nurse assistants, published in peer-reviewed journals, and written in English were included. Articles unrelated to the target population or concept were excluded, such as those focused on young populations or with the main focus on artificial nutrition challenges. No time limits were considered when performing the searches, as some insights were deemed relevant regardless of the publication year.

The two authors involved in this step adhered to the Preferred Reporting Items for Systematic Reviews and Meta-Analyses Extension for Scoping Reviews (PRISMA-ScR) guidelines to ensure a systematic and transparent process [36]. The selection process adhered to the PRISMA-ScR guidelines and is summarized in Figure 1.

A total of 4505 records were identified across four databases: PubMed (n = 1703), CINAHL (n = 650), WoS (n = 939), and Scopus (n = 1213). After removing 522 duplicate records, 3983 records remained for screening. During the title and abstract screening stage, 3948 records were excluded, including 68 studies focused on breastfeeding, 252 on artificial nutrition and/or hydration, 336 on young populations, and 3292 that were not pertinent to nutritional competencies or behaviors. This resulted in 35 records sought for full-text retrieval, all of which were successfully retrieved. Following full-text eligibility assessment, 17 additional records were excluded, including five focusing on artificial nutrition and/or hydration and 12 focusing on young populations. Ultimately, 18 studies met the inclusion criteria and were included in the review [27,37,38,39,40,41,42,43,44,45,46,47,48,49,50,51,52,53].

All included records were systematically examined using a pre-defined data extraction grid that was tailored to the review’s objectives. The extracted information was then subjected to a narrative synthesis, enabling an integrated interpretation of findings in alignment with the core research aims. The included articles were published between 1997 and 2022. Five were cross-sectional studies [37,40,42,43,46], three were randomized controlled trials (RCTs) [44,52,53], three were literature reviews [39,45,48], two were qualitative studies [38,47], two reported a mixed-method design [27,50], one was a quasi-experimental study [41], one was a discussion paper [49], and one was a Delphi study [51].

As this scoping review was not intended as stand-alone research but as a method-ological step for conceptualizing the new scale, the main content of each included study is summarized in Appendix A, while the summary of the included studies by study design is available in Appendix A, and the summary of the included studies by geographic location is presented in Appendix A.

#### 2.2.2. Phase 1b: Focus Group

The final results of the review were presented to the involved experts and discussed with the Italian research team during a focus group session (see Appendix A). The focus group was conducted as part of the conceptualization phase to refine the understanding of challenges, components, and practices related to nutritional care behaviors and to define themes that subsequently guided the development of the items. This step was conducted in December 2022. As per previous research [27], the focus group involved experts with demonstrated experience in nutritional care, specifically targeting research or clinical care for older adults. Participants were purposively selected by the two authors (RC and AM) based on the following criteria: (a) expertise in nutritional care either in research or in clinical practice, including care for older adults; (b) demonstrate involvement in developing tools or providing direct care for older populations; (c) holding advanced degrees relevant to their area of expertise (e.g., MSN, PhD). Eligible participants who were not proficient in speaking Italian were excluded from the focus group since the initial development of the items was in Italian.

A total of 10 experts participated in the focus group after purposively inviting them (100% response rate). The group comprised clinical nurses with expertise in nutritional care for older adults (70%) and three experts in tool development (30%). Three were males (30%), seven were female (70%), and the mean age was 37.3 years (standard deviation, SD = 6.5). Four hold a PhD (40%), and six had a Master of Nursing Science (MScN) (60%). The focus group session was conducted in a structured format, facilitated by two researchers. Participants discussed key findings from the scoping review, identifying the main challenges and opportunities in nutritional care for older adults. The session was audio-recorded, transcribed verbatim, and analyzed to extract themes and insights. The focus group discussions informed the refinement of the scale’s conceptual framework and the identification of specific dimensions and items for the initial item pool. The focus group lasted 95 min.

Following the thematic analysis [54], the first theme highlighted the abilities required to accurately assess patients’ nutritional status and calculate their specific dietary needs. The second theme emphasized evaluating nutritional information and developing individualized care plans tailored to the needs of patients. The third theme focused on the practical aspects of delivering nutritional care, including supporting and monitoring the effects of nutritional interventions on older adults. These three themes formed the conceptual foundation of the B-NNC Scale. Table 2 presents the scale blueprint that maps each thematic area to a corresponding scale domain. Specifically, the first theme informed the domain “Nutritional Assessment and Calculation Skills”, the second theme supported the domain “Nutritional Evaluation and Care Planning”, and the third theme guided the development of the domain “Nutritional Support and Care Implementation”. For each domain, representative items were derived to reflect specific behaviors grounded in the empirical literature and expert insights. This blueprint illustrates the coherence between the conceptualization and operationalization phases, ensuring that the scale structure reflects a systematic synthesis of existing evidence and clinical expertise.

#### 2.2.3. Phase 1c: Development of the Pool of Items

The development of the initial pool of items was guided by the findings from the scoping review and the themes gathered in the focus group discussions. This step was performed in January 2023 and led to the development of 24 items. Each item was designed to address key challenges, components, and practices related to nutritional care behaviors for older adults in clinical settings. These items aimed to reflect the practical actions registered nurses and nurse assistants performed in their day-to-day practice when caring for older adults at risk of or experiencing nutritional issues.

The items were conceptually anchored in the following prompt: “Thinking about your daily work with older patients who may have nutritional problems, please indicate how often you perform the following actions”. Participants will respond to the items using a 5-point Likert scale, with response options ranging from “Never” to “Daily”.

Table 3 lists the 24 items in Italian and provides an English translation for reporting purposes. The English translations are not intended for cross-cultural validation but rather to convey the conceptual content of the items.

### 2.3. Validation Phase (Phase 2)

#### 2.3.1. Phase 2a: Content Validity in Italy

To evaluate the content validity of the Italian version of the B-NNC Scale, a panel of 18 experts in nutritional care was purposively invited to participate in February 2023. These experts were asked to assess the relevance and essentiality of each item in the scale using the Content Validity Ratio (CVR) methodology [55,56]. The inclusion criteria for the experts were as follows: (a) demonstrated expertise in nutritional care, either through clinical practice, academic research, or both; (b) experience working with older adults or in contexts involving nutritional care interventions as a registered nurse; (c) a minimum of five years of professional experience; (d) holding advanced academic qualifications (e.g., MSN, PhD) or senior clinical roles related to nutritional care. Nonproficiency in Italian was the only exclusion criterion.

The CVR is a quantitative index used to evaluate the degree of agreement among experts on the essentiality of each item. Each panelist was asked to rate each item based on the following prompt: “Considering a scale aimed to measure nutritional care behaviors in older adults, how essential are the following items?” CVR values range from −1 to +1, with higher values indicating greater agreement on the essentiality of the item. The response options were essential, useful but not essential, not essential. The critical CVR value determines whether an item is retained or removed based on its agreement level. To establish the critical CVR, exact binomial probabilities were used [55,56], ensuring that the level of agreement exceeded that which might occur by chance.

The exact probability [*P*(*X ≥ n_e_*)] is calculated for each possible number of panelists who rate the item as essential (*n_e_*), where *X* is a random variable following a binomial distribution, *N* is the number of experts in a content validity assessment, *p* is the probability of success in a single assessment (assumed as *p* = 0.5 for equal likelihood of agreement or disagreement), $\left(\begin{array}{c} N\\ k \end{array}\right)$ is binomial coefficient, which calculates the number of ways to choose *k* successes (i.e., assessment of an item as essential) from *N* assessments:(1)$$P\left(X\geq n_{e}\right)=\overset{N}{\underset{k=n_{e}}{\sum }}\left(\begin{array}{c} N\\ k \end{array}\right)p^{k}{\left(1-p\right)}^{N-k}$$

For *N* = 18 and a significance level of 0.05, the critical VR is approximately 0.44, corresponding to at least 14 out of 18 panelists rating the item as “essential”. Therefore, items with a CVR value of 0.44 or higher were considered to have sufficient content validity and were retained in the scale. Items falling below this threshold were either revised or removed.

#### 2.3.2. Phase 2b: Translation and Adaptation Process in Austria

The translation and adaptation process for the Austrian version of the B-NNC Scale followed a rigorous methodology to ensure cultural and linguistic equivalence while maintaining the conceptual integrity of the original Italian version that reached adequate content validity [31,32,33]. This phase was performed between January 2023 and May 2023. The original Italian version of the B-NNC Scale was independently translated into German by two bilingual experts, both fluent in Italian and German and familiar with healthcare and nursing (forward translation). The consensus version of the Austrian translation was independently back-translated into Italian by two bilingual translators who were not involved in the forward translation process (back-translation). These translators had expertise in healthcare and nursing terminology. The consensus meeting involved all four translators (two forward translators and two back-translators), as well as the Austrian project leader, and aimed at comparing the original Italian version, the forward translation, and the back translation. Discrepancies in meaning, sentence structure, and terminology were discussed, and a pre-final version of the scale was agreed upon. The pre-final version of the Austrian B-NNC Scale was pilot-tested with 29 nurses who worked in clinical settings and had experience with nutritional care for older adults. Participants completed the behavior scale and assessed the understandability of the items and instructions using a dichotomous scale (clear or unclear). Items rated as unclear by at least 20% of participants were flagged for re-evaluation. The pilot test revealed a mean understandability rate of 97.8%, with individual item understandability rates ranging from 85.2% to 100%. Since all items surpassed the predetermined threshold for clarity, no changes were required to the final version of the scale.

#### 2.3.3. Phase 2c: Cross-Sectional Data Collection in Austria to Assess Psychometric Characteristics of the Austrian German Version of the Scale

A cross-sectional data collection was conducted between May and July 2023 to assess the psychometric characteristics of the German version of the B-NNC Scale. The study targeted nurses and nurse assistants employed in hospitals in Austria with more than 6 months of experience.

The invitation to take part in the study was distributed conveniently to large hospital organizations in Austria and via various social media platforms. The sample size for the cross-sectional study was calculated based on the need to perform a cross-validation procedure, which involved randomly splitting the sample into two subsets: one for Exploratory Factor Analysis (EFA) and the other for Confirmatory Factor Analysis (CFA). Approximately 55% of the sample was allocated to the EFA, while the remaining 45% was dedicated to the CFA [57]. Recommendations for factor analysis guided the sample size calculation by keeping into account the scale’s 24 items [58], which suggest an N:p ratio (participants per item) ranging from 10:1 to 25:1. This calculation indicated that 500 to 650 participants would be needed for each factor analysis (EFA and CFA), ensuring adequate statistical power. Therefore, a total target sample size of 1250 participants was deemed appropriate to allow for cross-validation, with sufficient sample sizes for both the EFA and CFA subsets.

Data collection was performed with LimeSurvey. The survey collected information on participants’ demographic and professional characteristics, including sex (male, female, I prefer not to respond), marital status (single, partnership, married, divorced, widowed), profession (nurse assistant, registered nurse), education level (non-university-based education, bachelor’s degree or equivalent, master’s degree, PhD), country of nursing education (educated in Austria, foreign educated in Europe, foreign educated outside Europe), work tenure (less than 5 years, 5 to 10 years, more than 10 years), clinical setting (surgical area, internal medicine, intensive care units, geriatric care, other clinical settings), tenure in the current workplace (less than 5 years, 5 to 10 years, more than 10 years), and working hours per week (up to 10 h/week, 11 to 20 h/week, 21 to 30 h/week, 31 or more h/week). Participants also reported their age and their level of engagement with assessing nutritional status in daily work (at least once per shift or less than once per shift).

#### 2.3.4. Phase 2d: Data Analysis to Assess Psychometric Characteristics of the German Version of the Scale

The collected data were summarized based on the nature of each variable and its distribution. Missingness in the dataset was always lower than 5%, and missing information was managed using multiple imputations under the Missing At Random (MAR) assumption to ensure the robustness of the analyses. Pairwise comparisons of the respondents’ characteristics were performed, considering the nature of the variable. For categorical variables, the chi-squared test was applied, while t-tests were used for normally distributed quantitative variables, and the Mann–Whitney test was applied for non-normally distributed variables. The Bonferroni correction was applied to account for multiple comparisons.

Prior to performing the EFA, the Kaiser–Meyer–Olkin (KMO) measure of sampling adequacy and Bartlett’s test of sphericity were conducted to evaluate the suitability of the data for factor analysis. The KMO statistic assessed the proportion of variance among the variables that could be explained by common factors, with values above 0.70 indicating acceptable adequacy. Bartlett’s test assessed whether the correlation matrix was significantly different from an identity matrix, ensuring the variables were correlated enough to proceed with the analysis. EFA was conducted on the first subsample (N = 602) in two rounds. The first round aimed to identify the most plausible factor structure and detect ambiguous items, defined as those showing cross-loadings with factor loadings equal to or higher than 0.32, which corresponds to 10% of the explained variance [57]. The EFA used a Maximum Likelihood Robust (MLR) estimation with 1000 iterations and a Geomin rotation. Several factor solutions ranging from two to four factors were evaluated based on scree plot interpretation. Parallel analysis was performed to confirm the factor structure, comparing the observed eigenvalues with those from random data generated using Monte Carlo simulations [59]. The settings for parallel analysis specified the number of cases, variables, and the desired factor analysis type.

The second round of EFA eliminated items with cross-loadings and tested the same factor solution to refine the structure. This process provided insights into the most plausible factor structure, which was subsequently used in the CFA.

CFA was performed on the second subsample (N = 470), testing a three-factor solution. The decision to perform CFA on the second subsample was based on the intent to cross-validate the factor structure identified through EFA in the first subsample. This split-sample approach enhances the methodological robustness of the validation process by testing how well the empirically derived structure fits an independent dataset. Model fit was evaluated using multiple fit indices, including Chi-Squared (χ^2^), Comparative Fit Index (CFI), Tucker–Lewis Index (TLI), Root Mean Square Error of Approximation (RMSEA) with a 90% confidence interval, and Standardized Root Mean Square Residual (SRMR). Chi-squared assessed the overall fit of the model, with smaller and non-significant values indicating a better fit. The CFI and TLI values above 0.90 were considered acceptable, while values above 0.95 indicated an excellent fit. An RMSEA value of 0.06 or less was interpreted as a good fit, and SRMR values of 0.08 or less were considered acceptable.

Measurement invariance was assessed across the full sample, including registered nurses (N = 917) and nurse assistants (N = 155), to evaluate whether the scale performed equivalently across these groups. The analysis tested configural invariance (basic factor structure consistency), metric invariance (equality of factor loadings), scalar invariance (equality of item intercepts), and strict invariance (equality of residual variances).

The reporting of estimates in all factor analyses included standardized values and their respective standard errors. Internal consistency was evaluated using McDonald’s Omega in both subsamples and the overall sample. Standardized scores for the overall sample and the two subgroups were calculated on a 0–100 scale. The scoring procedure involved summing the observed item responses for each domain, subtracting the minimum possible responses, and multiplying by 100/(maximum possible responses − minimum possible responses).

Finally, a correlational analysis was performed to describe the relationship between the scale scores and the respondents’ characteristics, providing additional insights into the association between the scale dimensions and demographic or professional variables. The data analysis was conducted using Mplus 8.1 (Muthén and Muthén), with a significance level of 5% and two-tailed tests.

## 3. Results

### 3.1. Content Validity of the Italian Version

A total of 18 nurse experts in clinical nutrition were enrolled to assess the content validity of the Italian version of the B-NNC Scale. Among the experts, there were six males (33.3%) and 12 females (66.7%), with a mean age of 37.56 years (SD = 6.68). The mean work tenure was 16.5 years (SD = 6.67). In terms of educational qualifications, the sample included 11 MScN graduates (61.1%), two PhD holders (11.1%), and two PhD students (11.1%).

All 24 items of the scale reported CVR higher than or equal to the critical value of 0.444, confirming their relevance. The CVR values ranged from 0.444 to 0.889, with a mean equal to 0.642 (SD = 0.11), as shown in Figure 2.

### 3.2. Validity of the German Version

The highest CVR (0.889) was observed for item 23, while the lowest CVR (0.444) was reported for items 2 and 12. This indicates adequate agreement among the experts regarding the importance of the scale items for measuring nursing nutritional care behaviors, allowing panelists to keep the items for subsequent phases.

#### 3.2.1. Phase 2: Sample Characteristics of the Cross-Sectional Data Collection in Austria Following the Translation and Adaptation Process

A total of 1072 nurses and nurse assistants participated in the cross-sectional data collection conducted in Austria, with the sample randomly divided into sub-group A (N = 602) for EFA and sub-group B (N = 470) for CFA. The demographic and professional characteristics of the overall sample and the two sub-groups are detailed in Table 4.

In the overall sample, 156 participants (14.6%) were male, 908 (84.7%) were female, and 8 (0.7%) preferred not to disclose their gender. The mean age of participants was 42.83 years (SD = 11.09), with a range from 22 to 65 years. Most participants were registered nurses (85.5%), while 14.5% were nurse assistants. In terms of educational background, 69.5% (n = 745) held an RN diploma only, 10.2% (n = 109) had a bachelor’s degree in nursing, 5.6% (n = 60) had a master’s degree, and 0.3% (n = 3) held a PhD.

Work tenure varied across the sample, with 17.4% having less than 5 years of experience, 16.9% between 5 and 10 years, and 65.8% more than 10 years. Clinical settings were diverse, with 22.1% working in internal medicine, 24.8% in geriatric care, 17.4% in surgical areas, 14.9% in intensive care units, and 20.8% in other clinical settings. Most participants worked full-time, with 67.2% reporting 31 or more working hours per week, while 1.3% reported working up to 10 hours per week.

Engagement with nutritional care in daily work was reported by 59.9% of participants as occurring at least once per shift, while 40.1% reported less frequent engagement. Sub-group comparisons between EFA and CFA samples revealed no statistically significant differences across most characteristics, also considering marital status (*p* = 0.024) and engagement with nutritional status in daily work (*p* = 0.045) because, after applying the Bonferroni correction for multiple comparisons, none of the observed differences reached the adjusted threshold for statistical significance (*p* ≤ 0.005).

#### 3.2.2. Phase 2: EFA in Sub-Group A

The initial analysis included all 24 items. Data were suitable for factor analysis (KMO = 0.963; Barlett’s test: χ^2^(276, N = 602) = 13,278.09, *p* < 0.001). Three different factor solutions (2-factor, 3-factor, and 4-factor) were evaluated based on statistical fit indices and interpretability (Table 5). The 2-factor solution served as the baseline model but demonstrated limited interpretability and lower fit indices (χ^2^(229, N = 602) = 1410.059, CFI = 0.875, TLI = 0.850, RMSEA = 0.093 [90% CI: 0.088–0.097], SRMR = 0.042). The 3-factor solution improved fit indices (χ^2^(229, N = 602) = 772.667, CFI = 0.940, TLI = 0.920, RMSEA = 0.060 [90% CI: 0.055–0.063], SRMR = 0.025) and showed better interpretability, with factors aligning to meaningful constructs. However, several items demonstrated cross-loadings exceeding 0.32, indicating ambiguity. The 4-factor solution (χ^2^(229, N = 602) = 671.729, CFI = 0.949, TLI = 0.924, RMSEA = 0.060 [90% CI: 0.054–0.063], SRMR = 0.022) did not significantly improve fit indices. The fourth factor failed to retain substantial item loadings, making it unsuitable for practical interpretation.

Five items with significant cross-loadings were removed to refine the factor structure, resulting in a 19-item scale. The refined analysis showed significant improvement in fit indices, particularly for the 3-factor solution (χ^2^(133, N = 602) = 446.391, CFI = 0.958, TLI = 0.940, RMSEA = 0.059 [90% CI: 0.056–0.065], SRMR = 0.024), with no cross-loadings observed. The 2-factor solution with the refined items (χ^2^(151, N = 602) = 1047.256, CFI = 0.879, TLI = 0.848, RMSEA = 0.099 [90% CI: 0.094–0.105], SRMR = 0.045) served as a baseline but lacked interpretability. The 4-factor solution with refined items (χ^2^(116, N = 602) = 640.065, CFI = 0.960, TLI = 0.950, RMSEA = 0.058 [90% CI: 0.056–0.068], SRMR = 0.022) failed to add meaningful value, with increased cross-loadings and an unstable fourth factor.

Table 6 provides the descriptive statistics for each item (mean, SD) and their factor loadings across the 2-factor, 3-factor, and 4-factor solutions in both rounds of analysis. The refined 3-factor solution demonstrated clear and strong item loadings without cross-loadings, confirming its robustness. The results from the parallel analysis strongly support the selection of a three-factor solution. The eigenvalues for the first three factors from the observed data exceeded the 95th percentile of eigenvalues derived from random datasets.

#### 3.2.3. Phase 2: CFA in Sub-Group B

The CFA was conducted on Sub-Group B (N = 470) to cross-validate the most plausible factor structure identified in the EFA phase. Based on the refined 19-item, three-factor structure, the model was tested for fit and interpretability. The three factors were labeled as Nutritional Assessment and Calculation Skills (F1), Nutritional Evaluation and Care Planning (F2), and Nutritional Support and Care Implementation (F3).

Table 7 summarizes the descriptive statistics and standardized factor loadings for each item in the model. All items demonstrated significant and strong loadings on their respective factors, further supporting the validity of the proposed structure. The model fit indices indicated that the structure provided a good representation of the data, with χ^2^(160, N = 470) = 417.100 = 417.100, CFI = 0.956, TLI = 0.947, RMSEA = 0.058 (90% CI: 0.052–0.065), and SRMR = 0.044.

These indices collectively confirm that the three-factor model is well-fitted and adequately explains the variance in the data. The factor loadings also align with the conceptual interpretation of each domain. F1 reflects evidence gathering, recognizing symptoms, and performing calculations for nutritional needs. F2 emphasizes evaluating habits, diagnosing issues, and planning interventions. F3 focuses on implementing practical interventions and supporting dietary needs during meals.

#### 3.2.4. Phase 2: Measurement Invariance Between Registered Nurses and Nurse Assistants in the Overall Sample

Measurement invariance between registered nurses (N = 917) and nurse assistants (N = 155) was assessed using the 19-item version of the scale. The analysis tested four levels of invariance: configural invariance, metric invariance, scalar invariance, and strict invariance, with results summarized in Table 8.

The configural invariance model demonstrated good fit indices (χ^2^(354, N = 1072) = 1210.326, CFI = 0.949, TLI = 0.944, RMSEA = 0.057 [90% CI: 0.055–0.069], SRMR = 0.058), indicating that the basic structure of the measurement model was equivalent across the two groups. This suggests that both registered nurses and nurse assistants shared a similar underlying factor structure for the measured constructs.

When testing metric invariance, which constrains factor loadings to be equal across groups, the fit indices slightly declined (χ^2^(359, N = 1072) = 1413.503, CFI = 0.916, TLI = 0.914, RMSEA = 0.068 [90% CI: 0.061–0.073], SRMR = 0.066). While the general structure was similar, this decline indicates some variability in how strongly items relate to the underlying factors across the two groups.

The scalar invariance model, which adds constraints on item intercepts to test if the groups interpret the items similarly, exhibited a more pronounced decline in fit (χ^2^(378, N = 1072) = 1951.820, CFI = 0.901, TLI = 0.899, RMSEA = 0.072 [90% CI: 0.063–0.089], SRMR = 0.088). This suggests potential differences in how the two groups perceive or respond to the items, likely influenced by differences in professional roles, responsibilities, or experiences between registered nurses and nurse assistants.

The strict invariance model, which adds constraints on residual variances, showed the poorest fit (χ^2^(417, N = 1072) = 2503.140, CFI = 0.876, TLI = 0.875, RMSEA = 0.094 [90% CI: 0.088–0.121], SRMR = 0.155). The significant decrease in fit suggests that residual variances differ between the groups, indicating that registered nurses and nurse assistants have distinct levels of response variability for some items.

#### 3.2.5. Phase 2: Reliability

The McDonald’s Omega values in Table 9 demonstrate high internal consistency across all factors and subgroups.

For F1 (Nutritional Assessment and Calculation Skills), the Omega values were 0.866 for Sub-group A, 0.864 for Sub-group B, and 0.865 for the overall sample. For F2 (Nutritional Evaluation and Care Planning), the Omega values were 0.940 for Sub-group A, 0.937 for Sub-group B, and 0.941 for the overall sample. For F3 (Nutritional Support and Care Implementation), the Omega values were 0.942 for both Sub-group A and Sub-group B, and 0.943 for the overall sample.

#### 3.2.6. Phase 2: Computed Scores

The scale scores were standardized on a 0–100 scale, with higher scores indicating better performance in each of the three identified factors: Nutritional Assessment and Calculation Skills (F1), Nutritional Evaluation and Care Planning (F2), and Nutritional Support and Care Implementation (F3). Table 10 summarizes the computed scores.

#### 3.2.7. Phase 2: Correlations

In the overall sample (Figure 3), Factor 1 (Nutritional Assessment and Calculation Skills) showed significant positive correlations with Factor 2 (Nutritional Evaluation and Care Planning) (r = 0.564, *p* < 0.001) and Factor 3 (Nutritional Support and Care Implementation) (r = 0.404, *p* < 0.001), indicating alignment among the factors in the scale.

It also demonstrated a weaker positive correlation with engagement in nutritional assessment during daily work (r = 0.123, *p* < 0.001). Age and tenure variables were weakly but significantly negatively correlated with Factor 1 (r = −0.086, *p* = 0.005; r = −0.094, *p* = 0.002). Factor 2 exhibited strong positive correlations with Factor 3 (r = 0.779, *p* < 0.001) and moderate positive correlations with engagement in nutritional assessment (r = 0.234, *p* < 0.001). Notably, it showed weak negative correlations with the professional role (r_pb_ = −0.108, *p* < 0.001: registered nurses exhibited lower scores) and work tenure (r = −0.011, *p* = 0.717). Factor 3 was strongly associated with Factor 2 (r = 0.779, *p* < 0.001) and moderately with engagement in nutritional assessment (r = 0.165, *p* < 0.001). It showed a positive association with work tenure (r = 0.095, *p* = 0.002) but no significant relationship with the profession or other demographic variables. In subgroup analyses (Figure 4), correlations revealed similar patterns.

## 4. Discussion

This study aimed to develop the B-NNC Scale in its original Italian version and translate and validate the German version among registered nurses and nurse assistants in Austria. In phase 1, the project successfully developed the B-NNC Scale based on an extensive literature review and input from experts in the field. The pre-final scale consists of 24 items. In phase 2, the scale’s content validity was found to be adequate, with a mean CVR of 0.634. The psychometric evaluation involved an EFA that suggested a 3-factor solution (“Nutritional Assessment and Calculation Skills”, “Nutritional Evaluation and Care Planning”, “Nutritional Support and Care Implementation”) and the exclusion of 5 items. The subsequent CFA confirmed this solution. Measurement invariance shows that registered nurses and nurse assistants have distinct levels of response variability for some items, and McDonald’s Omega demonstrated high internal consistency with the scale.

The content validity of an instrument is regarded as the most important psychometric property because instrument items have to be both relevant and comprehensive with respect to the construct of interest [60]. In the current study, an extensive scoping review built the basis for developing the B-NNC Scale, which exceeds commonly used standards for developing scales [60]. After the literature review, a focus group discussion was held, which guided the development of the items that were subsequently discussed by another panel of experts. This procedure underlines the efforts made by the authors to ensure good content validity.

Building on this foundation, the B-NNC Scale demonstrated robust psychometric properties in Austria. The EFA’s 3-factor solution captured distinct, meaningful dimensions of nutritional care behaviors, and the CFA further confirmed that this model fits the data well. Notably, McDonald’s Omega indicated high internal consistency across all subgroups and the overall sample, confirming the reliability of the scale. While measurement invariance tests suggested partial differences in how registered nurses and nurse assistants respond to certain items, the overall factor structure appears stable and applicable in a range of nursing contexts. This underscores the value of the B-NNC Scale as a flexible instrument that could be used in different professional groups, with caution exercised in direct comparisons between them.

Beyond the scale’s structural validity, insights into current nursing practice were gleaned from associations between the scale’s factors and participant characteristics such as age, tenure, and engagement with nutritional care. Particularly noteworthy is the finding that nurse assistants scored higher on Factor 2 (“Nutritional Evaluation and Care Planning”) compared to registered nurses, suggesting a potential gap in fundamental nursing care activities among RNs. From a clinical standpoint, this is concerning because RNs often hold broader responsibilities for care coordination [61]. Ensuring that fundamental nutritional care is consistently prioritized and integrated into practice may require targeted interventions, especially for registered nurses who face competing demands [62]. Future studies with larger samples are warranted to explore these correlates in multivariable models, thereby providing a more nuanced understanding of how demographic and professional factors influence nutritional care behaviors. Ultimately, such insights inform tailored educational programs and policies aimed at bolstering the delivery of high-quality nutritional care across diverse nursing roles.

The study also has some limitations. First, the psychometric validation of the B-NNC Scale was conducted exclusively in an Austrian sample. While the initial scale development and content validation were carried out in Italy, including a focus group and an expert panel of Italian-speaking professionals, the decision to perform full psychometric validation in Austria was based on practical considerations. Specifically, Austrian partner institutions were able to provide timely access to a sufficiently large and diverse sample of nurses and nurse assistants, which was essential for robust exploratory and confirmatory factor analyses and measurement invariance testing. At the time of data collection, no comparable sampling opportunity was feasible in Italy.

Nevertheless, this was one of the first studies to comprehensively develop and extensively psychometrically validate a scale to measure nurses’ nutritional care behaviors. Many studies in the field capture only several psychometric properties, but our study draws a comprehensive picture of the psychometric properties of the German B-NNC Scale [63,64,65,66]. The methodological approach was very rigorous and adhered to recognized standards and guidelines. The authors’ extensive experience in the development and testing of instruments has also contributed to this [19,20,27].

Moreover, while the B-NNC Scale was specifically designed for use by nurses and nurse assistants; its core behavioral domains, such as nutritional assessment, individualized planning, and implementation of supportive care, may be relevant to other healthcare providers and caregivers involved in geriatric nutritional management. However, the current version of the tool has not been tested in these broader professional groups. This limits the generalizability of the findings beyond the nursing profession. Future research is needed to evaluate the scale’s applicability across a broader range of healthcare roles and care settings, which could enhance its utility in interdisciplinary teams and long-term care environments.

Furthermore, while the current version of the B-NNC Scale addresses key behaviors related to protein intake and general caloric needs, it does not yet differentiate among macronutrient subtypes such as polyunsaturated fats, omega-3, and omega-6 fatty acids, or explicitly assess behaviors related to lipid and carbohydrate intake. These components are known to play a critical role in cardiovascular and cellular health, and their consideration is essential in promoting balanced nutritional care across different sociocultural and clinical settings. Future refinement of the scale may explore the integration of items targeting these specific aspects of dietary assessment and intervention, thereby enhancing its scientific and clinical value in supporting comprehensive nutritional management.

For further research, we recommend complementing this research with data from the context of Italian healthcare. This would allow a comparison to be drawn between Austria and Italy. The B-NNC Scale can be applied in practice to measure the behavior of nurses comprehensively, but it can also be used as a supplement to other related concepts like knowledge, attitudes, and self-efficacy. This would also extend the scientific knowledge regarding the association between these concepts and offer practical ideas to improve nutritional care in the long run, such as tailored educational programs.

## 5. Conclusions

This study successfully developed the B-NNC Scale in Italian and validated its German version among Austrian registered nurses and nurse assistants. The findings point to a robust, three-factor structure that captures essential behavioral domains of nutritional care, high internal consistency, and acceptable content validity. Although measurement invariance analyses revealed some differences in item responses between registered nurses and nurse assistants—particularly in the domain of fundamental nursing care—overall, the B-NNC Scale appears well-suited for diverse nursing contexts. Future research should expand on these results to further explore how demographic and professional factors influence nutritional care behaviors, ideally through larger and more complex analyses. The B-NNC Scale provides a critical means to assess and enhance the implementation of best practices in malnutrition management, ultimately supporting improved health outcomes for older adults.

## Figures and Tables

**Figure 1 nursrep-15-00146-f001:**
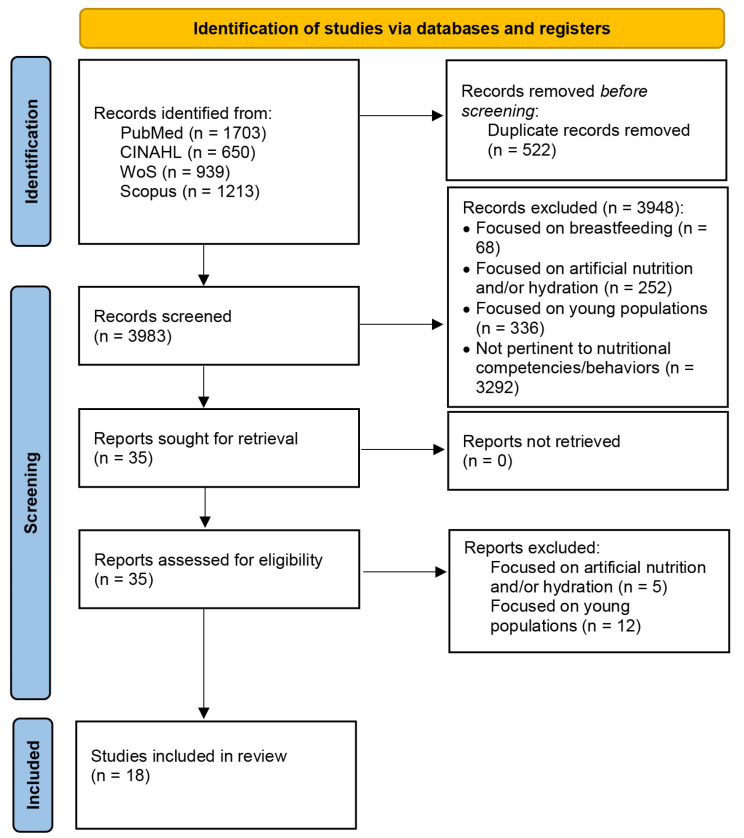
Selection flow diagram of the literature review performed in Phase 1.

**Figure 2 nursrep-15-00146-f002:**
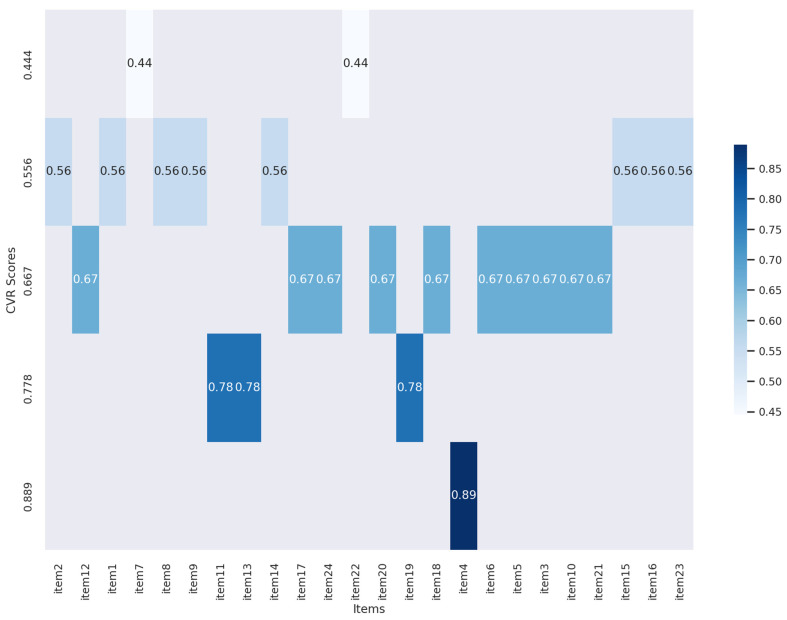
Heatmap of CVR scores.

**Figure 3 nursrep-15-00146-f003:**
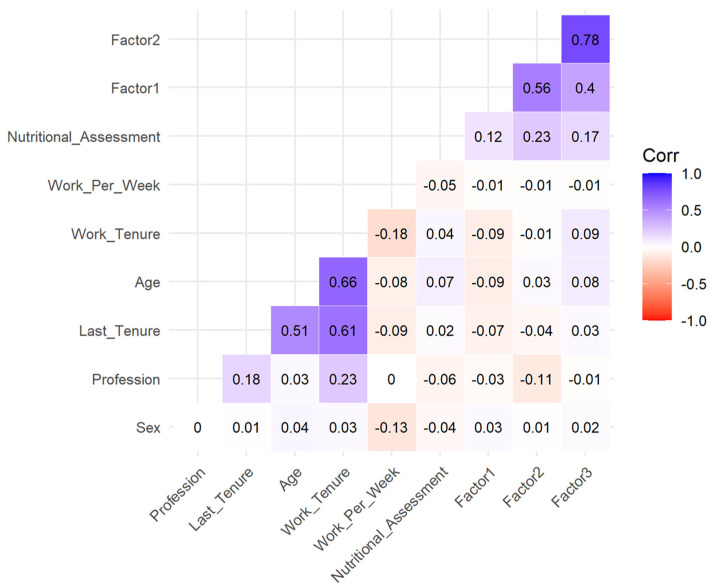
Correlogram for the overall sample.

**Figure 4 nursrep-15-00146-f004:**
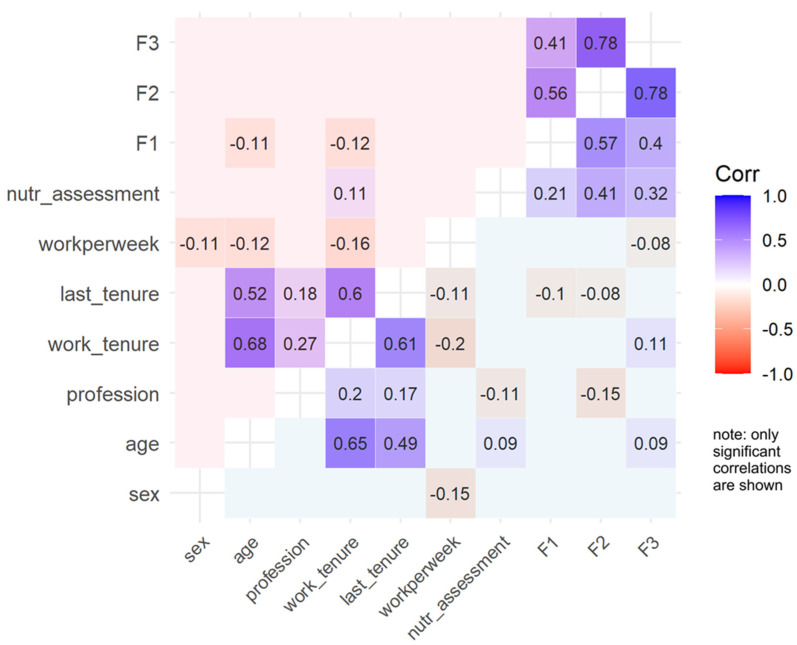
Correlogram in the two sub-groups. Note: Sub-group A has a blue background, and subgroup B has a pink background.

**Table 1 nursrep-15-00146-t001:** Search strategy.

Database	Final Query	Search Date	Records
PubMed	(Nurses[Mesh] OR nurs*[tiab] OR “nurse assistant*”[tiab] OR “nurs* aid*”[tiab]) AND ((behavior*[tiab] OR behaviour*[tiab]) AND (“Nutritional Status”[tiab] OR “nutritional care”[tiab] OR nutrition*[tiab]))	November 2022	1073
CINAHL	((MH Nurses+) OR (TI nurs* OR AB nurs*) OR (TI “nurse assistant*” OR AB “nurse assistant*”) OR (TI “nurs* aid*” OR AB “nurs* aid*”)) AND (((TI behavior* OR AB behavior*) OR (TI behaviour* OR AB behaviour*)) AND ((TI “Nutritional Status” OR AB “Nutritional Status”) OR (TI “nutritional care” OR AB “nutritional care”) OR (TI nutrition* OR AB nutrition*)))	November 2022	650
WoS	(ALL=Nurses OR (TI=nurs* OR AB=nurs*) OR (TI=“nurse assistant*” OR AB=“nurse assistant*”) OR (TI=“nurs* aid*” OR AB=“nurs* aid*”)) AND (((TI=behavior* OR AB=behavior*) OR (TI=behaviour* OR AB=behaviour*)) AND ((TI=“Nutritional Status” OR AB=“Nutritional Status”) OR (TI=“nutritional care” OR AB=“nutritional care”) OR (TI=nutrition* OR AB=nutrition*)))	November 2022	939
Scopus	(INDEXTERMS(Nurses) OR TITLE-ABS(nurs*) OR TITLE-ABS(“nurse assistant*”) OR TITLE-ABS(“nurs* aid*”)) AND ((TITLE-ABS(behavior*) OR TITLE-ABS(behaviour*)) AND (TITLE-ABS(“Nutritional Status”) OR TITLE-ABS(“nutritional care”) OR TITLE-ABS(nutrition*)))	November 2022	1213

**Table 2 nursrep-15-00146-t002:** B-NNC scale blueprint: Relationship between Phase 1 themes and scale theoretical domains.

Theme from Scoping Review and Focus Group	Final Scale Domain	Evidence Base	Representative Items
Theme 1: Assessment and calculation of nutritional needs. Identifying malnutrition, interpreting clinical data, and estimating dietary requirements.	F1. Nutritional Assessment and Calculation Skills	[27,37,41,44,50,53]	Estimate basal metabolic rate. Use validated tools to assess nutritional status. Interpret lab and anthropometric data.
Theme 2: Evaluation and planning of individualized nutritional care. Identifying preferences, evaluating behaviors, risk factors, and applying evidence in care planning.	F2. Nutritional Evaluation and Care Planning	[27,39,40,45,50,51,52]	Identify dietary preferences. Evaluate adherence to dietary recommendations. Adapt care plans using current guidelines.
Theme 3: Implementation of supportive nutritional care in daily practice. Providing hands-on support, organizing mealtime environments, and fostering intake.	F3. Nutritional Support and Care Implementation	[38,42,43,46,47,48,49]	Provide meal assistance. Organize meals to improve intake. Support feeding in dependent older adults.

**Table 3 nursrep-15-00146-t003:** Initial pool of items.

Item (Italian)	Item (English Conceptual Translation)
Reperire raccomandazioni e evidenze dalla letteratura relative alle alterazioni dello stato nutrizionale delle persone anziane	Find recommendations and evidence from literature on nutritional alterations in older people
2. Riconoscere segni e sintomi precoci di alterazione dello stato nutrizionale delle persone anziane	Recognize early signs and symptoms of altered nutritional status in older people
3. Stimare il metabolismo basale di una persona anziana	Estimate the basal metabolic rate of an older person
4. Stimare il fabbisogno energetico giornaliero totale delle persone anziane in funzione del peso corporeo, attività condotta e condizioni cliniche	Estimate the total daily energy requirements of older people based on body weight, activity, and clinical conditions
5. Interpretare i valori degli esami di laboratorio con valore predittivo di alterazione dello stato nutrizionale	Interpret laboratory findings predictive of altered nutritional status
6. Interpretare gli indici antropometrici e di trofismo muscolare	Interpret anthropometric and muscle parameters
7. Implementare strategie per favorire la relazione tra persona anziana e altri assistiti al fine di favorire il consumo del pasto	Implement strategies to foster relationships among older people to improve meal consumption
8. Implementare strategie per organizzare il momento del pasto in modo da favorire un maggior consumo calorico-proteico	Implement strategies to organize mealtime to promote higher caloric-protein intake
9. Adattare i piani assistenziali e la pratica quotidiana sulla base di raccomandazioni e evidenze disponibili in letteratura relative alle alterazioni nutrizionali	Adapt nursing plans and daily practices based on evidence and recommendations for nutritional alterations
10. Identificare e valutare le preferenze nutrizionali di ciascuna persona anziana	Identify and evaluate the dietary preferences of each older person
11. Identificare e valutare le persone anziane a rischio di alterazioni dello stato nutrizionale e/o con alterazioni conclamate	Identify and assess older people at risk of or experiencing nutritional alterations
12. Identificare e valutare i comportamenti alimentari delle persone anziane	Identify and evaluate the eating behaviors of older people
13. Identificare e valutare i fattori di rischio modificabili e non modificabili di un’alterazione dello stato nutrizionale delle persone anziane	Identify and assess modifiable and non-modifiable risk factors for altered nutritional status in older people
14. Identificare e valutare comprensione, conoscenze e stili di vita delle persone anziane	Identify and assess the understanding, knowledge, and lifestyle of older people
15. Valutare e monitorare nel tempo i fattori che possono influenzare gli esiti clinici nutrizionali delle persone anziane	Evaluate and monitor factors influencing clinical nutritional outcomes over time
16. Valutare e monitorare nel tempo i segni e i sintomi di stato nutrizionale alterato delle persone anziane	Evaluate and monitor signs and symptoms of altered nutritional status in older people over time
17. Valutare e monitorare nel tempo l’aderenza delle persone anziane alle raccomandazioni relative allo stile alimentare	Evaluate and monitor the adherence of older people to dietary recommendations
18. Identificare e valutare il setting appropriato per favorire l’assunzione del pasto	Identify and evaluate the appropriate setting to promote food intake
19. Valutare lo stato nutrizionale tramite strumenti validati	Evaluate nutritional status using validated tools
20. Garantire l’assunzione del fabbisogno calorico e proteico delle persone anziane	Ensure the caloric and protein intake of older people is adequate
21. Garantire l’erogazione dei pasti sulla base delle preferenze ed esigenze personali delle persone anziane	Provide meals based on the personal preferences and needs of older people
22. Fornire pasti adeguati sia in termini di qualità che di quantità	Provide meals of adequate quality and quantity
23. Supportare l’assunzione dell’adeguato apporto calorico nelle persone anziane impossibilitate a nutrirsi autonomamente	Support older people who cannot feed themselves independently
24. Organizzare il momento del pasto in modo da favorirne la completa assunzione	Organize mealtimes to promote full food intake

**Table 4 nursrep-15-00146-t004:** Demographic and professional characteristics in the overall sample and their comparisons between sub-group A (EFA) and sub-group B (CFA).

		Overall (N = 1072)		Sub-Group A (N = 602)		Sub-Group B (N = 470)		*p* *
		N	%	N	%	N	%	
Sex								
	Male	156	14.6	89	14.8	67	14.3	0.915
	Female	908	84.7	509	84.6	399	84.9	
	I prefer not to respond	8	0.7	4	0.7	4	0.9	
Marital status								
	Single	200	18.7	95	15.8	105	22.3	0.024
	Partnership	335	31.3	181	30.1	154	32.8	
	Married	453	42.3	276	45.8	177	37.7	
	Divorced	79	7.4	47	7.8	32	6.8	
	Widowed	5	0.5	3	0.5	2	0.4	
Profession								
	Nurse assistant	155	14.5	83	13.8	72	15.3	0.535
	Registered Nurse	917	85.5	519	86.2	398	84.7	
Education								
	Nurse assistant	155	14.5	83	13.8	72	15.3	0.880
	RN diploma only	745	69.5	422	70.1	323	68.7	
	RN Bachelor’s Science	109	10.2	62	10.3	47	10	
	Master’s Degree	60	5.6	34	5.6	26	5.5	
	PhD	3	0.3	1	0.2	2	0.4	
Work tenure								
	Less than 5 years	186	17.4	98	16.3	88	18.7	0.547
	5 to 10 years	181	16.9	101	16.8	80	17	
	More than 10 years	705	65.8	403	66.9	302	64.3	
Clinical setting								
	Surgical area	186	17.4	105	17.4	81	17.2	0.100
	Internal medicine	237	22.1	144	23.9	93	19.8	
	Intensive care units	160	14.9	94	15.6	66	14	
	Geriatric care	266	24.8	131	21.8	135	28.7	
	Other clinical settings	223	20.8	128	21.3	95	20.2	
Tenure in current workplace								
	Less than 5 years	436	40.7	232	38.5	204	43.4	0.267
	5 to 10 years	181	16.9	104	17.3	77	16.4	
	More than 10 years	455	42.4	266	44.2	189	40.2	
Working hours per week								
	Up to 10 h/week	14	1.3	5	0.8	9	1.9	0.340
	11 to 20 h/week	91	8.5	51	8.5	40	8.5	
	21 to 30 h/week	247	23.0	146	24.3	101	21.5	
	31 or more h/week	720	67.2	400	66.4	320	68.1	
Age								
	Years, range: 22–65 (mean; SD)	42.83	11.09	43.11	10.63	42.48	11.66	0.524
Engagement with nutritional status in daily work								
	At least once per shift	642	59.9	377	62.6	265	56.4	0.045
	Less than once per shift	430	40.1	225	37.4	205	43.6	

Note: In these comparisons (*), a significance level of *p* ≤ 0.005 was considered statistically significant. This threshold was adjusted using the Bonferroni correction to account for multiple comparisons and reduce the risk of inflating Type I errors. None of the comparisons showed statistically significant differences.

**Table 5 nursrep-15-00146-t005:** Fit indices and interpretability of factor solutions with original and refined item sets for exploratory factor analysis.

	Item	Chi-Squared	CFI	TLI	RMSEA [90% CI]	SRMR	Note
2-Factor Solution	original items (n = 24)	χ^2^(229, N = 602) = 1410.059	0.875	0.850	0.093 [0.088–0.097]	0.042	The 2-factor solution was used as the baseline model (reference point) in this analysis. Subsequent models were evaluated for improvements in statistical fit and interpretability.
3-Factor Solution	original items (n = 24)	χ^2^(229, N = 602) = 772.667	0.940	0.920	0.60 [0.055–0.63]	0.025	This solution shows adequate interpretability, with some cross-loadings, and a good fit in explaining sample statistics. This solution was also confirmed by suggesting the retention of 3 factors based on eigenvalues exceeding the 95th percentile random eigenvalues.
4-Factor Solution	original items (n = 24)	χ^2^(229, N = 602) = 671.729	0.949	0.924	0.60 [0.054–0.63]	0.022	The 4-factor solution does not improve the goodness of fit compared to solutions with fewer factors. The quantity of cross-loadings increases, making the solution unsuitable for practical interpretation. Furthermore, the 4th factor fails to retain any substantial item loading, indicating that it does not represent a meaningful or distinct construct within the data.
2-Factor Solution	after removing cross-leading (n = 19)	χ^2^(151, N = 602) = 1047.256	0.879	0.848	0.099 [0.094–0.105]	0.045	The 2-factor solution was used as the baseline model (reference point) in this analysis. Subsequent models were evaluated for improvements in statistical fit and interpretability.
3-Factor Solution	after removing cross-leading (n = 19)	χ^2^(133, N = 602) = 446.391	0.958	0.940	0.059 [0.056–0.065]	0.024	This solution shows adequate interpretability, no cross-loadings, and a good fit for sample statistics. This solution was also confirmed by suggesting the retention of 3 factors based on eigenvalues exceeding the 95th percentile random eigenvalues.
4-Factor Solution	after removing cross-leading (n = 19)	χ^2^(116, N = 602) = 640.065	0.960	0.950	0.058 [0.056–0.068]	0.022	The 4-factor solution does not improve the goodness of fit compared to solutions with fewer factors. The quantity of cross-loadings increases, making the solution unsuitable for practical interpretation. Furthermore, the 4th factor fails to retain any substantial item loading, indicating that it does not represent a meaningful or distinct construct within the data.

Legend: CFI = Comparative Fit Index; TLI = Tucker–Lewis Index; RMSEA = Root Mean Square Error of Approximation; SRMR = Standardized Root Mean Square Residual.

**Table 6 nursrep-15-00146-t006:** Descriptive statistics (N = 602) and factor loadings from exploratory factor analyses with 1-, 3-, and 4-factor solutions.

			2-Factor Solution		3-Factor Solution			4-Factor Solution			
Round 1. Original number of items											
Item	mean	SD	F1	F2	F1	F2	F3	F1	F2	F3	F4
Item1	4.02	0.969	0.217 (0.043)	**0.498 (0.047)**	**0.482 (0.048)**	0.198 (0.074)	0.045 (0.064)	**0.457 (0.071)**	0.196 (0.099)	0.074 (0.070)	−0.111 (0.097)
Item2	2.67	1.077	**0.711 (0.030)**	0.096 (0.042)	0.078 (0.035)	**0.516 (0.070)**	0.261 (0.072)	0.067 (0.040)	**0.556 (0.079)**	0.208 (0.085)	0.039 (0.070)
Item3	4.3	1.035	0.002 (0.010)	**0.915 (0.023)**	**0.934 (0.022)**	−0.017 (0.015)	0.021 (0.031)	**0.926 (0.023)**	−0.022 (0.024)	0.034 (0.026)	−0.004 (0.020)
Item4	4.27	1.057	−0.003 (0.011)	**0.908 (0.021)**	**0.896 (0.026)**	0.028 (0.025)	−0.015 (0.029)	**0.913 (0.024)**	0.031 (0.019)	−0.031 (0.024)	0.053 (0.031)
Item5	3.71	1.293	0.259 (0.043)	**0.367 (0.052)**	0.300 (0.050)	**0.417 (0.065)**	−0.104 (0.062)	0.297 (0.060)	**0.442 (0.075)**	−0.135 (0.068)	−0.004 (0.062)
Item6	3.52	1.281	**0.392 (0.040)**	0.288 (0.049)	0.246 (0.049)	**0.413 (0.072)**	0.031 (0.064)	0.227 (0.062)	**0.430 (0.083)**	0.023 (0.066)	−0.060 (0.072)
Item7	2.27	1.287	**0.873 (0.026)**	−0.110 (0.040)	−0.031 (0.022)	0.209 (0.078)	**0.704 (0.068)**	−0.057 (0.026)	0.207 (0.108)	**0.711 (0.097)**	−0.020 (0.059)
Item8	2.62	1.366	**0.856 (0.026)**	−0.078 (0.037)	0.010 (0.017)	0.159 (0.084)	**0.732 (0.071)**	−0.018 (0.027)	0.158 (0.113)	**0.742 (0.097)**	−0.027 (0.055)
Item9	3.42	1.373	**0.582 (0.034)**	0.250 (0.042)	0.217 (0.045)	**0.498 (0.068)**	0.148 (0.058)	0.175 (0.077)	**0.511 (0.109)**	0.169 (0.078)	−0.159 (0.101)
Item10	2.67	1.303	**0.873 (0.020)**	−0.034 (0.029)	−0.011 (0.024)	**0.459 (0.066)**	**0.474 (0.062)**	−0.049 (0.031)	**0.470 (0.110)**	**0.483 (0.097)**	−0.086 (0.083)
Item11	3.08	1.337	**0.690 (0.033)**	0.169 (0.042)	0.080 (0.037)	**0.800 (0.046)**	−0.010 (0.035)	0.050 (0.053)	**0.850 (0.060)**	−0.050 (0.059)	−0.078 (0.077)
Item12	2.85	1.302	**0.870 (0.015)**	−0.002 (0.009)	−0.025 (0.024)	**0.643 (0.057)**	0.308 (0.057)	−0.060 (0.033)	**0.669 (0.093)**	0.294 (0.089)	−0.077 (0.079)
Item13	2.91	1.318	**0.771 (0.033)**	0.104 (0.043)	0.003 (0.024)	**0.899 (0.028)**	−0.012 (0.034)	−0.004 (0.030)	**0.957 (0.049)**	−0.095 (0.058)	0.044 (0.049)
Item14	2.98	1.281	**0.709 (0.033)**	0.125 (0.038)	0.034 (0.027)	**0.818 (0.042)**	−0.005 (0.046)	0.027 (0.031)	**0.871 (0.056)**	−0.080 (0.059)	0.036 (0.045)
item15	2.72	1.345	**0.880 (0.021)**	−0.037 (0.033)	−0.081 (0.022)	**0.723 (0.063)**	**0.351 (0.061)**	−0.063 (0.030)	**0.845 (0.092)**	0.065 (0.050)	0.276 (0.073)
Item16	2.72	1.309	**0.859 (0.024)**	0.034 (0.035)	−0.004 (0.019)	**0.681 (0.059)**	**0.367 (0.062)**	0.026 (0.035)	**0.829 (0.102)**	0.043 (0.041)	**0.361 (0.108)**
Item17	2.99	1.332	**0.818 (0.025)**	0.040 (0.034)	0.015 (0.025)	**0.612 (0.063)**	0.281 (0.066)	0.012 (0.030)	**0.676 (0.064)**	0.189 (0.069)	0.103 (0.079)
Item18	2.92	1.341	**0.835 (0.024)**	0.025 (0.033)	0.062 (0.032)	**0.361 (0.076)**	**0.526 (0.067)**	0.039 (0.039)	**0.385 (0.088)**	**0.503 (0.081)**	−0.005 (0.071)
Item19	3.94	1.256	**0.370 (0.040)**	**0.410 (0.050)**	**0.358 (0.050)**	**0.446 (0.067)**	−0.020 (0.046)	0.316 (0.089)	0.449 (0.103)	0.020 (0.037)	−0.204 (0.110)
Item20	3.29	1.328	**0.648 (0.029)**	0.215 (0.038)	0.206 (0.039)	**0.429 (0.063)**	0.280 (0.055)	0.198 (0.044)	**0.459 (0.069)**	0.238 (0.066)	0.033 (0.068)
Item21	2.71	1.307	**0.828 (0.031)**	−0.037 (0.042)	0.089 (0.035)	−0.030 (0.048)	**0.887 (0.042)**	0.066 (0.032)	−0.044 (0.049)	**0.906 (0.039)**	0.000 (0.051)
Item22	2.82	1.337	**0.760 (0.032)**	0.054 (0.039)	0.160 (0.041)	0.023 (0.048)	**0.768 (0.053)**	0.148 (0.042)	0.020 (0.041)	**0.763 (0.054)**	0.047 (0.074)
Item23	2.34	1.355	**0.857 (0.029)**	−0.108 (0.042)	−0.018 (0.027)	0.151 (0.065)	**0.744 (0.059)**	−0.016 (0.032)	0.163 (0.074)	**0.702 (0.078)**	0.129 (0.093)
Item24	2.8	1.364	**0.790 (0.032)**	−0.022 (0.045)	0.095 (0.041)	−0.008 (0.036)	**0.824 (0.036)**	0.094 (0.041)	−0.014 (0.051)	**0.808 (0.057)**	0.104 (0.094)
Round 2. After removing cross-loadings for the most plausible solution (3-factor solution)											
	mean	SD	F1	F2	F1	F2	F3	F1	F2	F3	F4
Item1	4.02	0.969	0.220 (0.043)	**0.483 (0.048)**	**0.477 (0.048)**	0.191 (0.073)	0.040 (0.067)	**0.494 (0.050)**	0.212 (0.139)	0.093 (0.109)	−0.065 (0.060)
Item2	2.67	1.077	**0.719 (0.032)**	0.076 (0.042)	0.068 (0.036)	**0.518 (0.071)**	0.263 (0.074)	0.095 (0.036)	**0.478 (0.095)**	**0.369 (0.092)**	−0.085 (0.065)
Item3	4.3	1.035	0.006 (0.013)	**0.913 (0.023)**	**0.947 (0.019)**	−0.024 (0.025)	0.002 (0.008)	**0.915 (0.027)**	−0.031 (0.026)	−0.019 (0.022)	0.105 (0.071)
Item4	4.27	1.057	−0.004 (0.009)	**0.915 (0.019)**	**0.906 (0.022)**	0.021 (0.014)	−0.034 (0.028)	**0.894 (0.031)**	0.004 (0.027)	0.002 (0.021)	0.039 (0.080)
Item5	3.71	1.293	0.263 (0.045)	**0.358 (0.054)**	0.288 (0.052)	**0.420 (0.066)**	−0.104 (0.060)	0.287 (0.056)	−0.041 (0.132)	**0.402 (0.077)**	−0.028 (0.105)
Item6	3.52	1.281	**0.399 (0.042)**	0.271 (0.050)	0.237 (0.049)	**0.400 (0.076)**	0.044 (0.070)	0.249 (0.051)	0.177 (0.141)	**0.318 (0.100)**	−0.047 (0.086)
Item7	2.27	1.287	**0.896 (0.023)**	−0.142 (0.031)	−0.036 (0.023)	0.220 (0.075)	**0.699 (0.066)**	−0.020 (0.023)	**0.828 (0.094)**	0.016 (0.054)	0.073 (0.150)
Item8	2.62	1.366	**0.870 (0.022)**	−0.103 (0.028)	0.010 (0.019)	0.168 (0.077)	**0.718 (0.067)**	0.008 (0.025)	**0.727 (0.106)**	0.008 (0.054)	0.170 (0.148)
Item9	3.42	1.373	**0.589 (0.036)**	0.226 (0.045)	0.198 (0.045)	**0.506 (0.068)**	0.143 (0.060)	0.199 (0.047)	0.218 (0.098)	**0.433 (0.085)**	0.014 (0.062)
Item11	3.08	1.337	**0.693 (0.037)**	0.146 (0.048)	0.049 (0.041)	**0.821 (0.050)**	−0.013 (0.039)	0.028 (0.030)	−0.041 (0.064)	**0.836 (0.064)**	0.040 (0.054)
Item12	2.85	1.302	**0.865 (0.021)**	−0.018 (0.028)	−0.038 (0.028)	**0.655 (0.054)**	0.297 (0.054)	−0.043 (0.027)	**0.327 (0.113)**	**0.583 (0.068)**	0.049 (0.087)
Item13	2.91	1.318	**0.774 (0.037)**	0.085 (0.048)	−0.029 (0.032)	**0.932 (0.025)**	−0.019 (0.027)	−0.051 (0.025)	−0.036 (0.050)	**0.955 (0.050)**	0.015 (0.037)
Item14	2.98	1.281	**0.710 (0.036)**	0.107 (0.042)	0.008 (0.024)	**0.834 (0.040)**	−0.005 (0.042)	0.001 (0.027)	0.033 (0.071)	**0.815 (0.057)**	−0.010 (0.052)
Item17	2.99	1.332	**0.820 (0.025)**	0.018 (0.030)	0.006 (0.021)	**0.598 (0.063)**	0.294 (0.066)	0.013 (0.028)	**0.380 (0.109)**	**0.500 (0.079)**	0.010 (0.076)
Item20	3.29	1.328	**0.665 (0.029)**	0.187 (0.039)	0.189 (0.038)	**0.440 (0.063)**	0.278 (0.056)	0.148 (0.039)	0.107 (0.077)	**0.448 (0.073)**	0.223 (0.069)
Item21	2.71	1.307	**0.848 (0.028)**	−0.065 (0.032)	0.093 (0.037)	−0.020 (0.041)	**0.874 (0.038)**	−0.007 (0.016)	0.317 (0.158)	0.025 (0.041)	**0.643 (0.147)**
Item22	2.82	1.337	**0.789 (0.029)**	0.022 (0.032)	0.158 (0.041)	0.038 (0.055)	**0.760 (0.053)**	0.025 (0.034)	0.021 (0.025)	0.166 (0.090)	**0.789 (0.081)**
Item23	2.34	1.355	**0.880 (0.026)**	−0.135 (0.030)	−0.012 (0.021)	0.142 (0.063)	**0.755 (0.058)**	−0.031 (0.027)	**0.658 (0.188)**	0.038 (0.054)	0.231 (0.194)
Item24	2.8	1.364	**0.811 (0.029)**	−0.048 (0.035)	0.103 (0.041)	−0.014 (0.051)	**0.828 (0.057)**	0.041 (0.033)	0.479 (0.227)	−0.021 (0.059)	**0.434 (0.216)**

Legend: SE = standard error; SD = standard deviation. Note: F1 focuses on gathering data from the literature, recognizing signs and symptoms, and performing calculations to assess nutritional needs (label: Nutritional Assessment and Calculation Skills). F2 emphasizes recognizing and diagnosing nutritional issues, evaluating habits, and planning interventions tailored to older adults based on evidence (label: Nutritional Evaluation and Care Planning). F3 highlights implementing interventions, supporting older adults during meals, and ensuring practical dietary strategies are in place to improve nutrition and quality of life (label: Nutritional Support and Care Implementation).

**Table 7 nursrep-15-00146-t007:** Confirmatory factor analysis (N = 470) to cross-validate the most plausible factor solution.

	Mean	SD	Nutritional Assessment and Calculation Skills (F1)	Nutritional Evaluation and Care Planning (F2)	Nutritional Support and Care Implementation (F3)
Item1	4.04	0.974	0.724 (0.040)		
Item2	2.66	1.075		0.765 (0.022)	
Item3	4.29	1.053	0.757 (0.040)		
Item4	4.27	1.055	0.721 (0.043)		
Item5	3.7	1.315		0.456 (0.038)	
Item6	3.44	1.286		0.531 (0.036)	
Item7	2.25	1.3			0.858 (0.018)
Item8	2.58	1.35			0.840 (0.021)
Item9	3.4	1.375		0.723 (0.023)	
Item11	3.06	1.348		0.790 (0.021)	
Item12	2.83	1.319		0.855 (0.017)	
Item13	2.91	1.332		0.839 (0.018)	
Item14	2.94	1.302		0.784 (0.022)	
Item17	2.97	1.352		0.838 (0.018)	
Item20	3.3	1.34		0.769 (0.021)	
Item21	2.67	1.31			0.862 (0.019)
Item22	2.79	1.343			0.818 (0.021)
Item23	2.28	1.357			0.877 (0.016)
Item24	2.78	1.366			0.841 (0.019)

Note: The model well explained sample statistics: χ^2^(160, N = 470) = 417.100; CFI = 0.956; TLI = 0.947; RMSEA [90% CI] = 0.058[0.052–0.065]; SRMR = 0.044.

**Table 8 nursrep-15-00146-t008:** Measurement invariance analysis across registered nurses and nurse assistants.

Invariance	Item	Chi-Squared	CFI	TLI	RMSEA [90% CI]	SRMR	Note
Configural invariance	version with 19 items	χ^2^(354, N registered nurses = 917; N Nurse assistants = 155) = 1210.326	0.949	0.944	0.057 [0.055–0.069]	0.058	The basic structure of the measurement model is equivalent across the groups (registered nurses and nurse assistants).
Metric invariance	version with 19 items	χ^2^(359, N registered nurses = 917; N Nurse assistants = 155) = 1413.503	0.916	0.914	0.068 [0.061–0.073]	0.066	There is an acceptable loss in invariance when constraining factor loadings to be equal across groups. However, the fit is still adequate.
Scalar invariance	version with 19 items	χ^2^(378, N registered nurses = 917; N Nurse assistants = 155) = 1951.820	0.901	0.899	0.072 [0.063–0.089]	0.088	This model shows borderline fits, suggesting potential issues with invariance when item intercepts are constrained across groups.
Strict invariance	version with 19 items	χ^2^(417, N registered nurses = 917; N Nurse assistants = 155) = 2503.140	0.876	0.875	0.094 [0.088–0.121]	0.155	The model fit significantly decreases, indicating poor invariance when residual variances are constrained.

Note: The analysis of measurement invariance across registered nurses (N = 917) and nurse assistants (N = 155) reveals varying levels of fit and interpretability. The configural invariance model, which examines whether the basic structure of the measurement model is equivalent across groups, demonstrated good fit indices (CFI = 0.949, RMSEA = 0.057, SRMR = 0.058). This suggests that both groups share a similar underlying factor structure for the measured constructs. However, the fit indices slightly declined when testing metric invariance, which constrains factor loadings to be equal across groups. This indicates that, while the general structure is similar, there may be some variability in how strongly items relate to the underlying factors across the two groups. The relatively small sample size for nurse assistants compared to registered nurses may contribute to this variability, as smaller sample sizes can make it more challenging to achieve stable estimates. The scalar invariance model, which further constrains item intercepts to test if the groups interpret the items similarly, exhibited a more pronounced decline in fit. This suggests potential differences in how the two groups perceive or respond to the items, likely influenced by differences in professional roles, responsibilities, or experiences that may not align perfectly between registered nurses and nurse assistants. The strict invariance model, which adds constraints on residual variances, showed the poorest fit, indicating that residual variances differ significantly between the groups. This lack of fit could be due to the highly unbalanced sample sizes, where the much smaller nurse assistant group may not provide enough statistical power to achieve invariance. Additionally, differences in training, job expectations, and work environments between the two groups could further contribute to these discrepancies.

**Table 9 nursrep-15-00146-t009:** Internal consistency (McDonald’s Omega).

	Sub-Group A EFA (N = 602)	Sub-Group B CFA (N = 470)	Overall (N = 1072)
	Omega	Omega	Omega
Nutritional Assessment and Calculation Skills (F1)	0.866	0.864	0.865
Nutritional Evaluation and Care Planning (F2)	0.940	0.937	0.941
Nutritional Support and Care Implementation (F3)	0.942	0.942	0.943

**Table 10 nursrep-15-00146-t010:** Computed scores of the Nursing Nutritional Care Behaviors Scale in the two sub-groups.

		Overall		Sub-Group A (N = 602)		Sub-Group B (N = 470)		*p*
		Mean	SD	Mean	SD	Mean	SD	
Nutritional Assessment and Calculation Skills (F1)								
	Score, range: 0–100	79.93	22.31	79.98	22.41	79.91	22.24	0.952
Nutritional Evaluation and Care Planning (F2)								
	Score, range: 0–100	52.33	25.45	52.04	25.49	52.56	25.44	0.736
Nutritional Support and Care Implementation (F3)								
	Score, range: 0–100	40.87	29.89	40.29	29.85	41.31	29.34	0.579

## Data Availability

The datasets generated and/or analyzed during the current study are not publicly available due to ethical and legal restrictions designed to protect participant confidentiality. In accordance with the informed consent obtained from participants, as well as institutional and General Data Protection Regulation (GDPR) policies, the data contain sensitive information that cannot be openly shared. However, de-identified data may be made available from the corresponding author upon reasonable request, provided that the requesting party presents a methodologically sound proposal aligned with the purpose of the original research. Data access may be granted upon approval from the relevant ethics committees and/or institutional review boards, and requestors may be required to sign a data-sharing agreement to ensure compliance with confidentiality and data protection regulations.

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
