# Peer review of "Development of the Nursing Nutritional Care Behaviors Scale (B-NNC) in Italian and Psychometric Validation of Its German Translation in Austria"

_nursrep, 2025, doi:10.3390/nursrep15050146_

Round 1
Reviewer 1 Report
Comments and Suggestions for Authors
- The basic concept of the study was unclear. The study applied two step from Italian language to Germany. Then why did this scale need to be translated into Italian first? How about the scale validation in Italian language itself?
- There is no clear information of participants in the abstract section.
- Abstract section failed to describe overall phase in the method section, please make it clearer so the reader can conclude well the study report by read the abstract section.
- The gap of “Nursing Nutritional Care Behaviors” was presented. However the gap and the problem statement regarding the issue of “Nursing Nutritional Care Behaviors” in Italian and German were lacking. Why the study need to be conducted in thos location? Many countries might has similar problem on it.
- The method section might lead to a misunderstanding, particularly on how the phases was presented. Phase 1 might consist of some step, thus steps must be presented in the different code. For example: Phase 1 (a). Literature Review, Phase 2 (b). Focus group, and etc.
- The statement of “A pilot test with Austrian nurses and nurse assistants was conducted to refine the translated version, ensuring the understandability of the items. Lastly, a cross-sectional study was carried out in Austria to collect data for the psychometric validation of the scale, including its reliability and construct validity.” leading to a question. What is the exclusion and inclusion criteria to be involved to the study?
- All figure and table must provide the information of abbreviation, statistical symbols, and also the statistical analysis that is utilized to get the data result.
- I suggest to provide of the scale blueprint which pooled from the literature review. It make a harmony between phases of the study.
Author Response
Comment 1. The basic concept of the study was unclear. The study applied two step from Italian language to Germany. Then why did this scale need to be translated into Italian first? How about the scale validation in Italian language itself?
Response 1: We thank the Reviewer for this important comment, which allowed us to clarify the rationale behind the two-phase, cross-national design of our study. We agree that, in the original version, this aspect may not have been sufficiently articulated.
To clarify, the B-NNC Scale was not translated into Italian. Instead, it was originally developed in Italian during Phase 1 of the study. This decision was taken for practical and organizational reasons, as the majority of the research team was based in Italy and Italian-speaking, which allowed for the efficient coordination of the item development process, including the scoping review, expert focus group, and content validation. The German version was obtained through a formal cross-cultural adaptation process following the development of the Italian version, and it was psychometrically validated in Austria due to the availability of a sufficiently large and diverse sample.
To enhance clarity, we have added a dedicated subsection titled “2.1.1. Development and Validation Settings” under the Methods section. This subsection explains the roles of each country in the development and validation phases and justifies the sequence of steps taken in the study.
In addition, we have expanded the Limitations section to transparently acknowledge that: (a) the Italian version has not yet undergone full psychometric validation, (b) the initial expert input came only from Italian-speaking professionals, and (c) future research should include psychometric testing of the Italian version and assess measurement invariance between the two linguistic versions.
Comment 2: There is no clear information of participants in the abstract section.
Response 2: We thank the Reviewer for highlighting this omission. In response, we have revised the abstract to explicitly report key details about the participants involved in the study. Specifically, we now clarify that 1,072 nurses and nurse assistants working in Austrian hospitals participated in the psychometric validation, and that they were drawn from a range of clinical settings including internal medicine, surgery, geriatrics, and intensive care. These changes enhance the transparency of the study population and setting.
Comment 3: Abstract section failed to describe overall phase in the method section, please make it clearer so the reader can conclude well the study report by read the abstract section.
Response 3: We appreciate this insightful comment. In response, we have revised the Methods section of the abstract to more clearly reflect the two-phase design of the study. We now explicitly distinguish Phase 1 (Development Phase), which involved item generation, expert consultation, and content validation in Italy, and Phase 2 (Validation Phase), which included the German translation, pilot testing, and psychometric validation in Austria. This clearer description improves the coherence and readability of the abstract, ensuring readers can better understand the overall flow of the study and its methodological rigor.
Comment 4: The gap of “Nursing Nutritional Care Behaviors” was presented. However the gap and the problem statement regarding the issue of “Nursing Nutritional Care Behaviors” in Italian and German were lacking. Why the study need to be conducted in thos location? Many countries might has similar problem on it.
Response 4: We thank the Reviewer for this insightful observation. In response, we revised the Introduction to explicitly clarify why Italy and Austria were selected as the initial contexts for the development and validation of the B-NNC Scale. We now describe how both countries face structural and organizational barriers to integrating nutritional care in nursing practice, as evidenced by prior research on knowledge gaps, limited routine implementation, and suboptimal adherence to nutritional care protocols. While many countries indeed experience similar challenges, we chose Italy and Austria based on the availability of collaborative research networks, linguistic accessibility, and practical conditions that allowed for robust scale development and psychometric testing.
The revised paragraph (starting with “In light of this global gap...”) provides this justification and situates our study within a broader international perspective, while explaining the methodological and operational rationale for starting with these two countries. We believe this enhancement improves the clarity and relevance of the study’s scope as per your request.
Comment 5: The method section might lead to a misunderstanding, particularly on how the phases was presented. Phase 1 might consist of some step, thus steps must be presented in the different code. For example: Phase 1 (a). Literature Review, Phase 2 (b). Focus group, and etc.
Response 5: We thank the Reviewer for this valuable suggestion. To improve clarity and ensure a more structured presentation of the study design, we have revised the Methods section by reformatting the subheadings. Each phase now includes clearly labeled and consistently coded sub-steps (e.g., Phase 1a: Literature Review; Phase 1b: Focus Group; Phase 2a: Content Validity Assessment; etc.). This revised structure ensures that readers can easily follow the sequence and organization of methodological procedures across both phases of the study.
Comment 6: The statement of “A pilot test with Austrian nurses and nurse assistants was conducted to refine the translated version, ensuring the understandability of the items. Lastly, a cross-sectional study was carried out in Austria to collect data for the psychometric validation of the scale, including its reliability and construct validity.” leading to a question. What is the exclusion and inclusion criteria to be involved to the study?
Response 6: We thank the Reviewer for this important observation. In response, we have revised the Methods section to clearly describe the inclusion and exclusion criteria applied in both Italy (content validity phase) and Austria (pilot test and cross-sectional validation). For the Italian phase, experts were included if they had at least five years of experience, expertise in clinical nutrition for older adults, and advanced academic qualifications; non-Italian speakers were excluded. For the Austrian phase, inclusion criteria comprised being a registered nurse or nurse assistant, currently employed in a hospital setting, and having at least six months of clinical experience. Individuals without direct patient care responsibilities or those still in training were excluded. These additions improve the transparency and replicability of the study.
Comment 7: All figure and table must provide the information of abbreviation, statistical symbols, and also the statistical analysis that is utilized to get the data result.
Response 7: We thank the Reviewer for this important technical suggestion. In response, we have carefully reviewed all figures and tables and revised their legends or footnotes.
Comment 8: I suggest to provide of the scale blueprint which pooled from the literature review. It make a harmony between phases of the study.
Response 8: We thank the Reviewer for this insightful suggestion. In response, we have added a scale blueprint (now presented as Table 2) that illustrates how the themes identified in the scoping review and focus group were systematically mapped onto the three theoretical domains of the B-NNC Scale. This table clearly links the conceptual findings from Phase 1 to the structure of the final scale. Additionally, we included a new explanatory paragraph in Section 2.2.2 to describe the blueprint and reinforce the alignment between the conceptualization and development phases.
Reviewer 2 Report
Comments and Suggestions for Authors
The article primarily illustrates how the authors developed a geriatric nutritional assessment tool for clinical nursing care. The study is highly significant, yet there are several aspects that require further clarification.
- The initial tool was developed in Italian, then translated into German and subsequently tested in Austria to arrive at the final version. While the authors may have had valid reasons for this approach, it is not immediately clear to readers. Typically, research tools are developed and finalized within the same language context. The rationale behind this cross-linguistic and cross-cultural adaptation process should be better explained to enhance the study's comprehensibility.
- The final tool is divided into three factors: Nutritional Assessment and Calculation Skills, Nutritional Evaluation and Care Planning, and Nutritional Support and Care Implementation. However, in most tool development processes, broad categories are usually established early on, followed by refinement of specific items. The article currently emphasizes the initial 24 items before arriving at these broader categories through factor analysis. The authors should elaborate on their rationale for this approach.
- The article contains excessive detail on procedural aspects, while the research background could benefit from a clearer focus on the key issues that the tool aims to address. For instance, the importance and necessity of the three factors—Nutritional Assessment and Calculation Skills, Nutritional Evaluation and Care Planning, and Nutritional Support and Care Implementation—should be better explained in the context of existing gaps in geriatric nutritional care.
Author Response
Comment 1: The article primarily illustrates how the authors developed a geriatric nutritional assessment tool for clinical nursing care. The study is highly significant, yet there are several aspects that require further clarification.
The initial tool was developed in Italian, then translated into German and subsequently tested in Austria to arrive at the final version. While the authors may have had valid reasons for this approach, it is not immediately clear to readers. Typically, research tools are developed and finalized within the same language context. The rationale behind this cross-linguistic and cross-cultural adaptation process should be better explained to enhance the study's comprehensibility.
Response 1: We thank the Reviewer for this important observation, which is fully aligned with a similar comment raised by Reviewer 1. To address this point, we have clarified the rationale for the two-country, cross-linguistic approach by revising the final paragraph of the Introduction and by adding a dedicated subsection in the Methods titled “2.1.1. Development and Validation Settings.” This section explains the practical and collaborative reasons for developing the original scale in Italian and conducting the psychometric validation in Austria. It also highlights how this design reflects the composition of the research team, the linguistic context, and access to adequate participant samples. We believe these revisions enhance the transparency and comprehensibility of the study’s design.
Comment 2:
- The final tool is divided into three factors: Nutritional Assessment and Calculation Skills, Nutritional Evaluation and Care Planning, and Nutritional Support and Care Implementation. However, in most tool development processes, broad categories are usually established early on, followed by refinement of specific items. The article currently emphasizes the initial 24 items before arriving at these broader categories through factor analysis. The authors should elaborate on their rationale for this approach.
Response 2: We thank the Reviewer for this insightful comment. To improve the clarity of our development process, we have added a scale blueprint (Table 2) and an accompanying paragraph in the Methods section (Section 2.2.2). This blueprint illustrates how the initial item pool was not constructed in an isolated or purely inductive manner, but was conceptually grounded in the themes that emerged from the scoping review and expert focus group conducted during Phase 1. These themes, corresponding to the ability to assess nutritional needs, plan and evaluate care, and implement support strategies, directly informed the item generation process. The factor analysis in Phase 2 empirically confirmed the conceptual structure, resulting in the final three domains. This approach is coherent with best practices in scale development as indicated in the foundational methodological studies cited in the design of this research. We believe this addition triggered by your comment demonstrates the coherence between conceptual development and statistical validation.
Comment 3:
- The article contains excessive detail on procedural aspects, while the research background could benefit from a clearer focus on the key issues that the tool aims to address. For instance, the importance and necessity of the three factors—Nutritional Assessment and Calculation Skills, Nutritional Evaluation and Care Planning, and Nutritional Support and Care Implementation—should be better explained in the context of existing gaps in geriatric nutritional care.
Response 3: We appreciate the Reviewer’s valuable feedback. In response, we have revised the Introduction to more clearly articulate the specific gap in the literature regarding the lack of validated tools assessing actual nursing behaviors in geriatric nutritional care. This reinforces the rationale for developing a behavior-focused scale, as opposed to existing tools that emphasize knowledge and attitudes. Furthermore, we clarified that the three domains of the final scale are not post hoc labels, but reflect key themes identified through the scoping review and expert focus group during Phase 1. As detailed in the revised Methods section (specifically in the scale blueprint and the new paragraph Section 2.2.2), the factor structure derived through EFA and CFA confirms these theoretically grounded domains. These enhancements ensure that the conceptual and empirical foundations of the three domains are clearly and coherently presented, addressing the Reviewer’s concern.
Reviewer 3 Report
Comments and Suggestions for Authors
This is a strong manuscript with the potential to make a significant impact in elder care.
Strengths:
This article provides a valuable contribution by introducing a tool designed to assess malnutrition among older adults, specifically tailored for nursing practice. The scale shows promise in improving early identification and intervention, which is crucial in geriatric care. The methodological rigor in scale development and the clarity of presentation are commendable.
Suggestions:
-
Target User Scope: While the tool is described as intended for nurses, its potential usefulness extends beyond nursing. Expanding the target audience to include a broader range of healthcare providers and caregivers could increase its utility and applicability in diverse settings. The authors are encouraged to address this point in the Discussion section and consider reframing the language to reflect a more inclusive approach.
-
Justification for CFA Use: The choice to apply Confirmatory Factor Analysis (CFA) for subgroup B is unclear. Since the development of the tool is not guided by an established theoretical framework, Exploratory Factor Analysis (EFA) would typically be more appropriate in early-stage scale development. The authors should either provide a clear justification for using CFA or reconsider their analytic approach, and explain this decision more thoroughly in the Methods or Results section.
Author Response
Comment 1: This is a strong manuscript with the potential to make a significant impact in elder care.
Strengths:
This article provides a valuable contribution by introducing a tool designed to assess malnutrition among older adults, specifically tailored for nursing practice. The scale shows promise in improving early identification and intervention, which is crucial in geriatric care. The methodological rigor in scale development and the clarity of presentation are commendable.
Response 1: Thank you for highlighting the strengths of our study.
Comment 2:
Target User Scope: While the tool is described as intended for nurses, its potential usefulness extends beyond nursing. Expanding the target audience to include a broader range of healthcare providers and caregivers could increase its utility and applicability in diverse settings. The authors are encouraged to address this point in the Discussion section and consider reframing the language to reflect a more inclusive approach.
Response 2: We thank the Reviewer for this thoughtful suggestion. In response, we have added a paragraph to the Discussion section, acknowledging that while the B-NNC Scale was developed and validated with nurses and nurse assistants, its behavioral domains may also be relevant to other healthcare providers involved in geriatric nutritional care. However, we also highlight that the current version of the scale has not been tested in other professional groups, and we present this as a limitation. We recommend that future studies explore the scale’s applicability in interdisciplinary teams and diverse care settings to assess its broader utility.
Comment 3: Justification for CFA Use: The choice to apply Confirmatory Factor Analysis (CFA) for subgroup B is unclear. Since the development of the tool is not guided by an established theoretical framework, Exploratory Factor Analysis (EFA) would typically be more appropriate in early-stage scale development. The authors should either provide a clear justification for using CFA or reconsider their analytic approach, and explain this decision more thoroughly in the Methods or Results section.
Response 3: We thank the Reviewer for raising this important point. In response, we have clarified our analytic rationale in Section 2.3.4. The decision to perform CFA on the second subsample was based on the intent to cross-validate the factor structure identified through EFA in the first subsample. This split-sample approach enhances the methodological robustness of the validation process by testing how well the empirically derived structure fits an independent dataset.
Reviewer 4 Report
Comments and Suggestions for Authors
I recommend that the authors describe, if they have identified the daily dietary intake of polyunsaturated fats, omega 6 and omega 3, not just the dietary intake of proteins. These fats have an increased functional importance with an impact on the health of the cardiovascular system and on the health of all cell membranes in the human body.
The study would have a greater scientific value presenting all three caloric nutrients, proteins, lipids and carbohydrates, which have an important significance in nutritional studies regarding total caloric intake and basal energy requirements in order to ensure and promote balanced nutrition regardless of the socio-cultural environment.
Author Response
Comment: I recommend that the authors describe, if they have identified the daily dietary intake of polyunsaturated fats, omega 6 and omega 3, not just the dietary intake of proteins. These fats have an increased functional importance with an impact on the health of the cardiovascular system and on the health of all cell membranes in the human body.
The study would have a greater scientific value presenting all three caloric nutrients, proteins, lipids and carbohydrates, which have an important significance in nutritional studies regarding total caloric intake and basal energy requirements in order to ensure and promote balanced nutrition regardless of the socio-cultural environment.
Response: We thank the Reviewer for this thoughtful observation. While the current version of the B-NNC Scale includes items that address energy and protein intake, it does not yet explicitly differentiate among macronutrient subtypes such as polyunsaturated fats (including omega-3 and omega-6 fatty acids) or carbohydrates. We have acknowledged this point in the Discussion and Limitations section, and we agree that these elements are functionally important to comprehensive nutritional care. We recommend that future refinement of the scale consider integrating behaviors related to the assessment and promotion of balanced intake across all major nutrient groups, which would further increase the tool’s clinical and scientific utility.
Round 2
Reviewer 2 Report
Comments and Suggestions for Authors
The author has provided a reasonable response to my questions, and I have no further comments. Given the significant value of this study in the field of geriatric nutritional care, I recommend its publication.